# Penta- and hexa-coordinated beryllium and phosphorus in high-pressure modifications of $CaBe_2P_2O_8$

Anna Pakhomova [1], Georgios Aprilis [2], Maxim Bykov [3], Liudmila Gorelova[4], Sergey S. Krivovichev[4,5], Maxim P. Belov[6], Igor A. Abrikosov[7] & Leonid Dubrovinsky [3]

Beryllium oxides have been extensively studied due to their unique chemical properties and important technological applications. Typically, in inorganic compounds beryllium is tetrahedrally coordinated by oxygen atoms. Herein based on results of in situ single crystal X-ray diffraction studies and ab initio calculations we report on the high-pressure behavior of $CaBe_2P_2O_8$, to the best of our knowledge the first compound showing a step-wise transition of Be coordination from tetrahedral (4) to octahedral (6) through trigonal bipyramidal (5). It is remarkable that the same transformation route is observed for phosphorus. Our theoretical analysis suggests that the sequence of structural transitions of $CaBe_2P_2O_8$ is associated with the electronic transformation from predominantly molecular orbitals at low pressure to the state with overlapping electronic clouds of anions orbitals.

[1] Deutsches Elektronen-Synchrotron (DESY), 22607 Hamburg, Germany. [2] Materials Physics and Technology at Extreme Conditions Laboratory of Crystallography, University of Bayreuth, 95440 Bayreuth, Germany. [3] Bayerisches Geoinstitut, University of Bayreuth, 95440 Bayreuth, Germany. [4] Institute of Earth Sciences, Saint-Petersburg State University, 199155 Petersburg, Russia. [5] Kola Science Center, Russian Academy of Sciences, Fersmana 14, 184209 Apatity, Russia. [6] Materials Modeling and Development Laboratory, NUST "MISIS", 119991 Moscow, Russia. [7] Department of Physics, Chemistry and Biology, Linköping University, 58183 Linköping, Sweden. Correspondence and requests for materials should be addressed to A.P. (email: anna.pakhomova@desy.de)

Due to the broad technological applications of beryllium oxocompounds[1–3], their structure and chemical bonding became a focus of a number of recent experimental and theoretical studies. Small atomic radius and high ionization energy of beryllium make covalent interactions playing the important role in stabilizing beryllium compounds, unlike other alkaline-earth elements. In contrast to MgO, CaO, SrO, and BaO, at ambient conditions BeO crystallizes in the hexagonal wurtzite structure, featuring tetrahedrally coordinated $Be^{2+}$ and $O^{2-}$ ions[4]. Bonding of beryllium to four oxygen atoms with the formation of $BeO_4^{6-}$ tetrahedra is also exclusive for its natural occurrence[5]. Various molecular oxygen-rich composition compounds ($BeO_2$, $Be_2O_2$, $Be_2O_4$, $BeO_4$, and $Be(O_3)_2$) demonstrating diverse Be–O bonding situations have been recently isolated within solid noble-gas matrices[6]. Cases with Be in coordination higher than four have not been observed experimentally for inorganic compounds, though recent ab initio calculation studies on $BeO_2$[7] and $BeO$[8] have predicted the formation of $BeO_6$ octahedra with ionic Be–O bonding at high pressures.

Recent advances at third-generation synchrotron facilities made the in situ high-pressure X-ray diffraction a powerful tool in simultaneous synthesis and structural characterization of new compounds[9–11]. High-pressure can effectively overcome reaction energy barriers and reorder atomic orbital energy levels so that new phases could reveal properties and stoichiometries unexpected from the viewpoint of conventional solid-state chemistry. Pressure-induced densification of matter is accompanied by the rearrangement of atomic bonds and structural units in order to fill the available space as effective as possible, which usually results in the increase of the atomic coordination numbers (see, e.g., reports on five- and six-fold coordination of silicon in glasses and melts[12–16] as well as in a number of crystalline silicates[17–23]).

Recent discovery of five- and six-fold coordinated silicon in the high-pressure phases of $CaB_2Si_2O_8$[19] has inspired us to probe high-pressure behavior of structurally similar compound $CaBe_2P_2O_8$. The question whether beryllium could experience the same increase in coordination number is of general chemical interest as well as of particular importance for the understanding the nature of Be–O and P–O bonding. While the first experimental observations of P[V] and P[VI] have been recently reported for high-pressure phases of $TiPO_4$[9] and $AlPO_4$[24], the current study appears to the first reporting on the experimental observation of five- and six-fold coordinated beryllium. Here, we present results of high-pressure single-crystal X-ray diffraction (SCXRD) experiments conjoined with ab initio density functional theory (DFT) calculations that evidence a step-wise transition of Be and P coordination from tetrahedral to octahedral through trigonal bipyramidal.

## Results

### Pressure-induced formation of hurlbutite-II with Be[V].

At ambient conditions hurlbutite, $CaP_2Be_2O_8$, possesses a monoclinic symmetry (space group $P2_1/c$) with $a = 7.798(3)$, $b = 8.782(2)$, $c = 8.299(1)$ Å, $\beta = 90.50(5)°$[25]. The asymmetric unit of hurlbutite contains four tetrahedrally coordinated T cations (two P and two Be), eight oxygen and one calcium atoms. Polymerization of $PO_4$ and $BeO_4$ tetrahedra through common vertices results in the formation of framework with four- and eight-membered channels running along the $a$-axis (Fig. 1). The eight-membered rings are occupied by Ca atoms that are seven-fold coordinated (for Ca–O bonds shorter than 3 Å).

Conventional continuous contraction of unit-cell parameters and atomic bonds is observed up to 7.5 GPa (Fig. 2). The compression of the unit cell is anisotropic so that the $c$-axis is the most and the $a$ axis is the least compressible (Supplementary

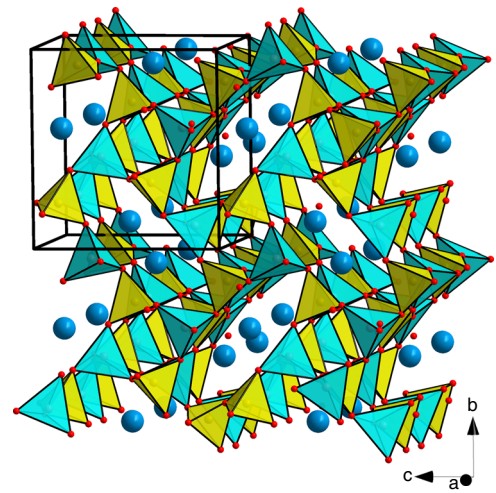

**Fig. 1** Crystal structure of hurlbutite, $CaBe_2P_2O_8$, at ambient conditions[25]. $BeO_4$ and $PO_4$ tetrahedra are given in blue and yellow, respectively. Ca and O atoms are given as blue and red spheres, respectively. Black solid line outlines a unit cell

Fig. 1). In agreement with earlier reports on the compression of similar frameworks[26], the compression of hurlbutite is controlled up to 7.5 GPa by changes in T–O–T angles of the tetrahedral framework, while the $TO_4$ tetrahedra stay as rigid units.

The unexpected response of the crystal structure on pressure treatment is observed above 7.5 GPa. While the $b$ and $c$ axes continue to decrease, the $a$-axis reveals an anomalous increase upon compression indicating a change in the compression mechanism (Fig. 2, Supplementary Fig. 1). Indeed half of the $TO_4$ units start to undergo pressure-induced geometrical distortion. The progressive deviation of $P1O_4$ and $Be2O_4$ tetrahedra from the ideal tetrahedral geometry above ~7.5 GPa is perfectly visible on plots showing quadratic elongation and bond angular variance parameters as a function of pressure[27] (Supplementary Fig. 2). Such a distortion results from the closure of eight-membered rings and progressive approach of the fifth oxygen to the P1 and Be2 atoms across the rings (Supplementary Fig. 3).

At pressures above 20 GPa the crystal structure again experiences conventional contraction with the preservation of the most ($c$-axis) and least ($a$-axis) compressible directions. The evolution of the new-forming contacts P1–O8* and Be2–O2* is shown in Supplementary Fig. 4. The shortening of the Be2–O2* contact is smooth however distribution of Be–O bonds indicates that the Be[IV] to Be[V] transition occurs between 70 and 75 GPa (Supplementary Tables 1 and 2). At 75 GPa $BeO_5$ polyhedra possesses trigonal bipyramidal geometry with two long apical (1.70 and 1.92 Å) and three short equatorial bonds (1.50–1.53 Å), while P1 still preserves tetrahedral coordination. Upon further compression of hurlbutite-II, $BeO_5$ evolves towards more regular trigonal bipyramid geometry by pronounced shortening of the Be–O2* bond.

### Crystal structure of hurlbutite-III with Be[V] and P[V].

At ~83 GPa the crystal structure undergoes another phase transition, which is reflected in an abrupt change of the unit-cell parameters (Fig. 2, Supplementary Fig. 1). The high-pressure phase hurlbutite-III preserves initial $P2_1/c$ symmetry. The phase transition to hurlbutite-III is displacive and induced by the final incorporation of O8* atoms into the coordination sphere of P1. At 83.2 GPa polyhedra of penta-coordinated P1 and Be2 possess

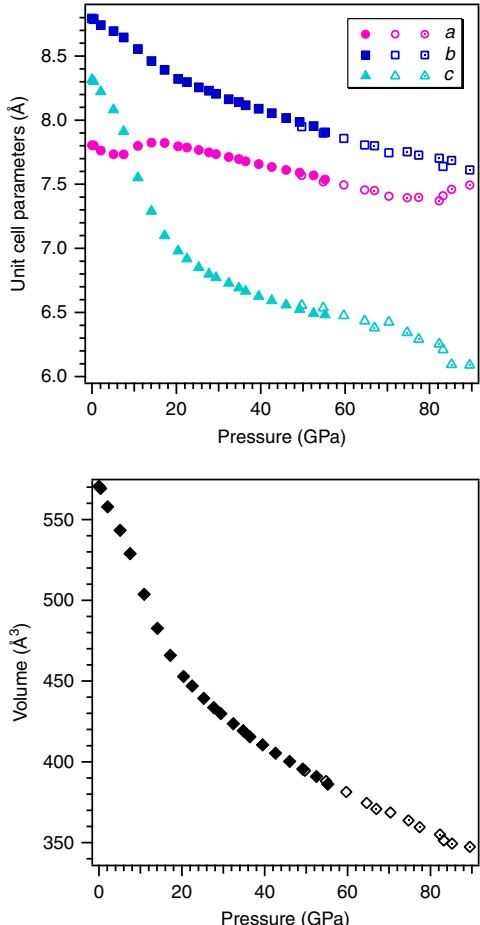

**Fig. 2** The evolution of the unit-cell parameters of hurlbutite, CaBe$_2$P$_2$O$_8$, along the compression. Filled, open and open with dots symbols correspond to high-pressure room-temperature experiments **1**–**3**, respectively. The errors are smaller than the size of the symbols

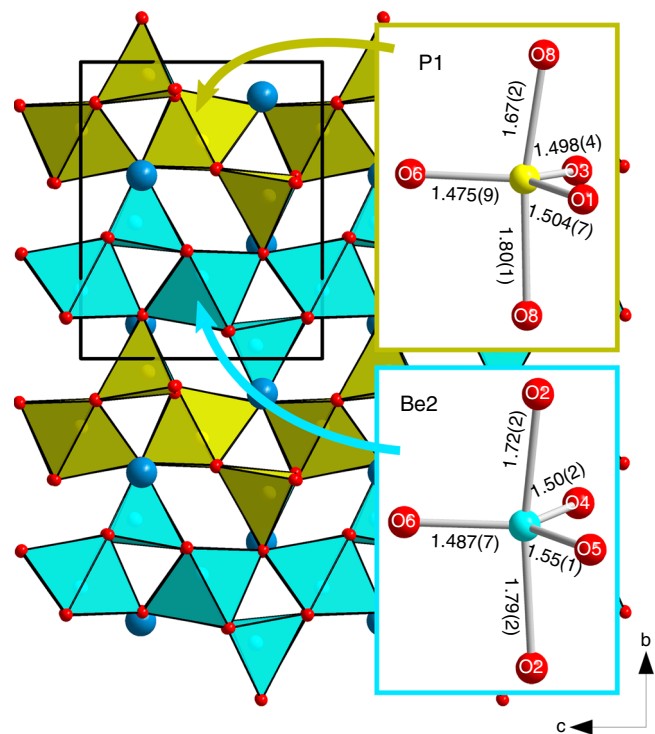

**Fig. 3** Crystal structure of hurlbutite-III, CaP$_2$Be$_2$O$_8$ at 83.2 GPa. BeO$_n$ and PO$_n$ polyhedra are given in blue and yellow. Ca and O atoms are given as blue and red spheres, respectively. Black solid line outlines a unit cell. Insets represent the trigonal bipyramidal geometry of PO$_5$ and BeO$_5$ groups

trigonal bipyramidal geometry (Fig. 3) with two long apical (1.662–1.813 Å for P1 and 1.711–1.798 Å for Be2) and three short equatorial bonds (1.471–1.506 Å for P1 and 1.492–1.545 Å for Be2; Supplementary Table 2). The O–T–O apical bond angles are 10.5° and 11.5° away from the 180° required for the regular trigonal bipyramid for P1 and Be2, respectively. The P2 and Be1 atoms remain tetrahedrally coordinated up to 82 GPa, so that the framework of hurlbutite-III is built upon TO$_4$ and TO$_5$ polyhedra sharing common vertices. The structure of hurlbutite-III contains distorted elements of both hexagonal and cubic packings (HCP and CCP, respectively), where the P1 and Be2 atoms fill trigonal bipyramidal voids and the P2 and Be1 atoms fill tetrahedral voids. Ca has 11-fold coordination manifesting distortion from an ideal close packing.

**Crystal structure of hurlbutite-IV with Be[VI] and P[VI].** Upon the compression above 90 GPa, new reflections appear in the diffraction patterns indicating the occurrence of another phase, coexisting with hurlbutite-III. The structure of this new phase, named hurlbutite-IV, was solved and refined in the $P$-1 space group (Supplementary Table 1). The asymmetric unit of hurlbutite-IV contains four crystallographically independent P, four Be, two Ca and 16 O atoms. The $P2_1/c \rightarrow P$-1 phase transition is reconstructive in character and involves rearrangement of the bonding network with the accompanying increase of coordination numbers for all cations. At 89.5 GPa, all P atoms are

octahedrally coordinated with the distribution of P–O bond distances varying in the range 1.49-1.77 Å (Supplementary Table 3). Be–O bond distances vary in the range 1.50–1.92 Å for Be1, Be2, and Be3 atoms that are octahedrally coordinated as well. The Be4 atom has five neighboring oxygens within the distance of 1.50–1.81 Å while the sixth Be4–O7 contact is of 2.09 Å. This distribution indicates that the coordination polyhedron of Be4 should better be described as a square pyramid. The geometry of individual P and Be polyhedra is shown in Fig. 4. The dense structure of hurlbutite-IV is built on the TO$_6$ and TO$_5$ polyhedra sharing common edges. The Ca and O atoms form distorted by stacking faults CCP arrangement, where P and Be atoms are filling octahedral voids and Ca atoms are 12-fold coordinated (Supplementary Fig. 5).

**Ab initio calculations of the transformation route.** In order to check whether experimentally observed behavior of CaP$_2$Be$_2$O$_8$ may be reproduced by theory we have performed ab initio simulations (See Methods) and found an excellent agreement between measured and calculated unit-cell parameters, volume, atomic coordinates and interatomic distances (Supplementary Figs. 6 and 7) as a function of pressure. Calculated pressure dependence of the interatomic distances (Supplementary Fig. 7) and enthalpy of high-pressure phases of hurlbutite (Supplementary Fig. 8) reproduce the experimentally observed sequence of transitions. According to the ab initio simulations, hurlbutite transforms to hurlbutite-II at 58 GPa, while hurlbutite-II to hurlbutite-III at calculated transition pressure of 67.5 GPa (slight underestimation of the transition pressure is typical for DFT calculations employing semilocal functionals, like GGA in our case, see Methods). Interestingly, these two transitions occur without any barrier as a result of relaxation of atomic positions in the unit cell. These three phases correspond to the topologically same enthalpy minimum that evolves with pressure smoothly but

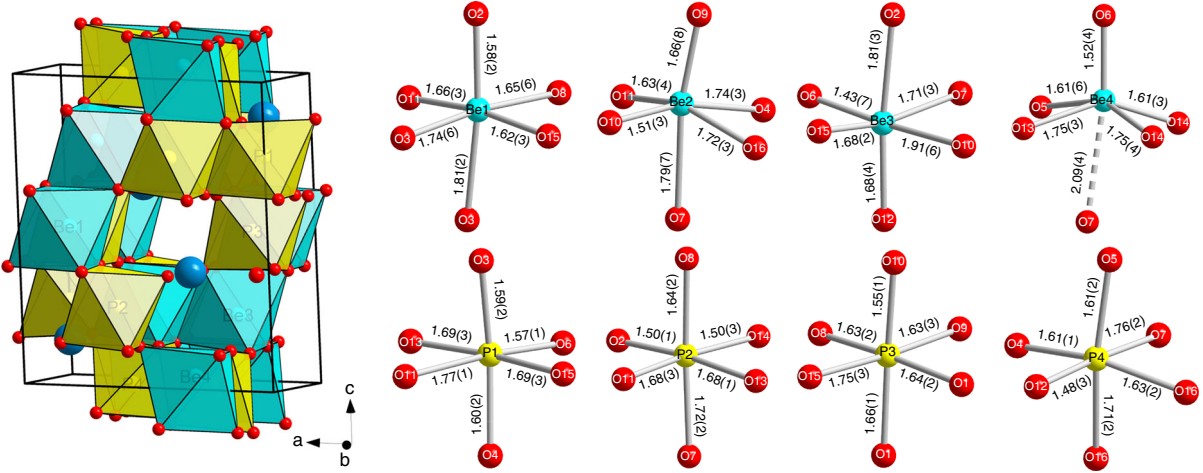

**Fig. 4** The crystal structure of hurlbutite-IV at 89.5(1) GPa. $PO_6$ octahedra are given in yellow, $BeO_n$ polyhedra are given in light blue. Ca and O atoms are presented as blue and red spheres

in a way that qualitatively modifies the local coordination of atoms in the unit cell at the corresponding transition pressures. On the contrary, calculated atomic configuration of hurlbutite-IV has its own enthalpy minimum, and the transition to this phase is of the first order, in agreement with experimental observation of coexistence of the phases III and IV. In fact, according to calculations hurlbutite-IV is the most thermodynamically stable phase at pressures above 65 GPa. Even though this pressure is most probably somewhat underestimated, it is very close to calculated transition pressure from hurlbutite-II to hurlbutite-III. Thus, theoretical analysis indicates that experimentally observed hurlbutite-III is likely a metastable phase of $CaBe_2P_2O_8$.

## Discussion

To the best of our knowledge, hurlbutite-II, -III, and -IV are the first examples of experimentally observed inorganic compounds possessing beryllium in coordination higher than four. Moreover, it appears to be the first structural report on an element of the second period manifesting such increase of coordination number. In a framework of classical Pauling's model[28] the $sp^3d$ and $sp^3d^2$ hybridization and, respectively, five- and six-fold coordination would not be expected for beryllium due to the absence of $d$ atomic orbitals. Despite the advances in the quantum chemistry in the second half of XX century and, particularly, introduction of a multicenter bonding model[29–31], there are still recent reports questioning whether the absence of $d$ orbitals prohibits the increased coordination of the elements of the second period[32].

Close agreement between our experimental observations and ab initio calculations make us confident that our theoretical approach should also adequately describe driving forces behind pressure-induced transformations in $CaP_2Be_2O_8$ as well as reasons for formation of phases with unusual coordinations of phosphorus and beryllium. In order to gain this knowledge, we have investigated the behavior of the electronic structure of $CaP_2Be_2O_8$ upon compression. Figure 5 summarizes calculated total electronic density of states (DOS) of different phases of hurlbutite. Analysis of the partial local DOS (Supplementary Fig. 9) demonstrates that all the occupied states of $CaP_2Be_2O_8$ are of $s$- and $p$-character: the electronic states corresponding to $d$ orbitals are well above the highest occupied state at all the studied pressures, and therefore are not relevant for the phase transitions observed in this work. At the same time, one can clearly see in Fig. 5a that the electronic structure of hurlbutite is characterized by well localized electronic states (peaks of the DOS are sharp and separated from each other by energy gaps). Upon compression, the states become broader,

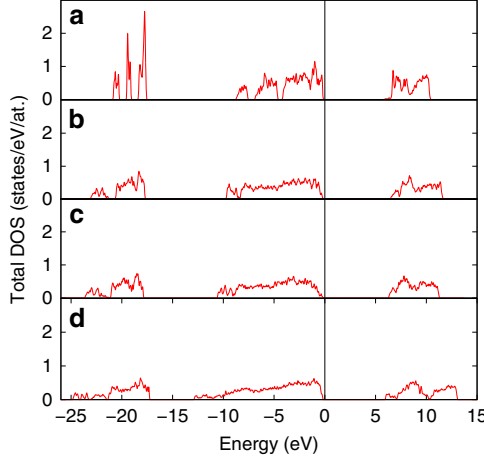

**Fig. 5** Calculated total electronic density of states (DOS) of different phases of hurlbutite, $CaP_2Be_2O_8$. **a** Hurlbutite at 2.9 GPa; **b** hurlbutite-II at 66.9 GPa; **c** hurlbutite-III at 68.7 GPa; **d** hurlbutite-IV at 79.6 GPa. Energy is given relative to the energy of the highest occupied state

and the energy gaps between them disappear, starting from the high-energy part of the spectrum in hurlbutite-II, and proceeding all the way to the low-energy part of the spectrum in hurlbutite-IV. We, therefore, conclude that the sequence of structural transitions observed in the present study is associated with the electronic transitions from predominantly molecular orbitals at low pressure to the state with overlapping electronic clouds of anions orbitals. Both experimental observations of the $BeO_5/BeO_6$ configurations and ab initio calculations are in line with previous quantum chemical calculations[33–36] and demonstrate that the involvement of $d$ orbitals is not mandatory for the formation of species with trigonal–bipyramidal and octahedral geometries. Instead, an electron-deficient multicenter bonding can be proposed as a mechanism of formation of such exotic configurations and, generally, as a densification mechanism for the $CaP_2Be_2O_8$ crystal structure adopting to high-pressure conditions[37,38].

Pressure-induced increase of cation coordination number is repeatedly observed tendency[38–40] that was outlined as a general rule of high-pressure crystal chemistry in reviews of Prewitt and Downs[41] and Grochala et al. (2007)[27]. In inorganic oxo-compounds this tendency is typically realized along with evolution of the oxygen sublattice toward the close packing arrangements[41]. This is also the case for hurlbutite: upon

pressurizing O and Ca cations firstly arrange into close packing assembly consisting both of distorted HCP and CCP elements (with P and Be occupying tetrahedral and trigonal bipyramidal voids in hurlbutite-II and -III) and later into ABCA-CABCBCAB arrangement (with P and Be filling octahedral voids in hurlbutite-IV). While the cation coordination and oxygen sublattice are traditional determinants of a crystal structure, the alternative view on the crystal chemistry of oxides was proposed by O'Keeffe and Hyde by introducing terms cation packing and anion coordination[42–44]. While this approach was found to be effective for a number of cases (e.g., elaborating on Al coordination in $Al^{VI}_2O_3$ and $Al^{IV}PO_4$, describing olivine → spinel transition), we find that the classical way of structure representation is more appropriate for our cases. By studying high-pressure behavior of a group of isotopological compounds with general formulae $MT1_2T2_2O_8$ (M = Ca, Ba, Sr; T = Si, B, Be, P), we conclude that the crystal structure response (and particularly formation of $TO_5$ species) is governed by interplay of two factors: compressibility of $TO_4$ tetrahedra and size of large M cation. The chemistry of the T site governs T[IV] → T[V] transition pressure (e.g., compare Be[V] and P[V] between 70–75 and 82–85 GPa in $CaBe_2P_2O_8$, respectively, and Si[V] at c.a. 22 GPa in $CaB_2Si_2O_8$[19]). The size of the M cation is responsible for formation of close packing arrangement with trigonal bipyramidal voids, i.e., $TO_5$ species. Thus, in contrast to $CaB_2Si_2O_8$, Si site in the structure of $SrB_2Si_2O_8$ does not change coordination to five-fold upon pressurizing but instead undergoes a splitting into two sites[45]. Further increase of M cation size results in complete absence of Si[V]: the high-pressure polymorphism is realized via direct Si[IV] → Si[VI] transition in the crystal structure of $BaB_2Si_2O_8$[45]. In the upcoming review article on high-pressure behavior of $MT1_2T2_2O_8$ compounds we are going to elaborate on the crystal chemical regularities in detail.

The present study further proves the powerful capabilities of high pressure as a tool for tuning chemical properties of matter. Growing interest of the chemical community in high-pressure SCXRD techniques using diamond anvil cells (DACs) ensures that the upcoming studies will bring further examples of unique phases as well as provide a solid experimental basis for the future development of novel high-pressure crystal chemistry.

## Methods

**High-pressure SCXRD experiments.** The natural samples of hurlbutite, $CaBe_2P_2O_8$, originating from Viitaniemi pegmatite (Eräjärvi area, Orivesi, Western and Inner Finland Region, Finland[46]) have been provided by Mineralogical Museum, CeNak, University of Hamburg. Three separate in situ high-pressure SCXRD were performed at the experimental station P02.2 (extreme conditions beamline) at synchrotron Petra III (Hamburg, Germany). Symmetric DACs with culets diameter of 300, 200, and 150 μm were used for pressure generation in experiments #1, #2, and #3, respectively. The sample chambers with approximate diameters of 150, 100, and 85 μm were obtained by drilling the preindented rhenium gasket. Hurlbutite crystals were placed inside the sample chambers along with a ruby sphere for pressure estimation (Supplementary Fig. 10)[47]. The DACs were loaded with neon as pressure-transmitting medium using the in-house gas loading system at Petra III. Monochromatic X-ray diffraction experiments were performed using X-rays with wavelength of ~0.2905 Å. The X-ray beam was focused to less than $10 \times 10$ μm by Kirkpatrick–Baez mirrors for experiments #1 and #3 and by Compound Reflective Lenses for experiment #2[48]. Diffraction patterns were collected using Perkin Elmer detector. Before each experiment the detector-sample distance was calibrated with a $CeO_2$ standard using the procedure implemented in the program Dioptas[49].

At each pressure both a wide-scan and a stepped ω-scan were collected for each crystal. Wide scans consisted of 40 s exposures during rotations of ±20° of the DAC. Step scans consisted of individual exposures taken over 0.5° intervals to constrain the ω angle of maximum intensity of each peak. Collected diffraction images were analyzed using the program CrysAlis Pro[©50]. The SHELXL program package was used for all structural determinations[51]. The high-pressure structural behavior of hurlbutite has been followed up to 91 GPa by performing SCXRD experiments at every pressure step of 3–5 GPa. In total 26 high-pressure structural refinements have been performed, the representative ones are given in Supplementary Tables 1 and 2.

**Ab initio DFT calculations.** Calculations of the electronic structure, total energies and structural parameters for all phases of hurlbutite were performed in the framework of the DFT using the Vienna ab initio simulation package[52,53]. The interaction between ions and electrons were described using the projector augmented-wave method[54,55] through recommended potentials for Ca, Be, P and O. A plane-wave basis set cutoff energy was set to 600 eV. The generalized gradient approximation (GGA) in the Perdew–Burke–Ernzerhof (PBE)[56] parametrization was selected to treat the exchange and correlation effects. PBE-GGA belongs to the so-called semilocal DFT functionals, which are known to slightly underestimate bonding strength in solids and significantly underestimate band gaps in semiconductors and insulators. However, it is numerically efficient and provides accuracy, which is sufficient for the purposes of the present study. The integration over the Brillouin zone was performed using a gamma-centered mesh with a k-points grid $6 \times 6 \times 6$. All the calculations were carried out using the Gaussian smearing with a broadening of $\sigma = 0.05$ eV. All properties were calculated at a given volumes (pressures) after the ion positions and cell shape were relaxed to achieve forces action on ions smaller then $10^{-2}$ eV/Å and hydrostatic conditions: $|p - p_x|$, $|p - p_y|$, $|p - p_z| < 0.5$ kbar.

## Data availability

The X-ray crystallographic coordinates for structures reported in this article have been deposited at the Inorganic Crystal Structure Database (ICSD) under deposition numbers CSD 1913366-1913370. These data can be obtained from CCDC's and FIZ Karlsruhe's free service for viewing and retrieving structures (https://www.ccdc.cam.ac.uk/structures/).

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

## Acknowledgements

Prof. Dr. Jochen Schlüter (Mineralogisches Museum, Centrum für Naturkunde (CeNak), Universität Hamburg, Grindelallee 48, 20146 Hamburg) is highly acknowledged for providing the samples. This research was carried out at the light source PETRA III at DESY, a member of the Helmholtz Association (HGF). Theoretical calculations of structural properties were supported by the Ministry of Science and High Education of the Russian Federation in the framework of Increase Competitiveness Program of NUST "MISIS" (No. K2-2019-001) implemented by a governmental decree dated 16 March 2013, No. 211. Theoretical analysis of the electronic structure was supported by the Russian Foundation for Basic Research (Grant 19-02-00871). Financial support from the Swedish Research Council (VR) through Grant no. 2015-04391 and the Swedish Government Strategic Research Areas in Materials Science on Functional Materials at Linköping University (Faculty Grant SFO-Mat-LiU No. 2009-00971) and SeRC are gratefully acknowledged.

## Author contributions

A.P., G.A., M. Byk, and L.G. conducted the high-pressure single-crystal X-ray diffraction experiments. A.P. analyzed the X-ray diffraction data. M. Bel. and I.A. conducted ab initio calculations. A.P., L.D., M. Bel, I.A. and S.K. interpreted the results and wrote the paper with the contribution of all authors.

## Additional information

**Competing interests:** The authors declare no competing interests.

