## [Peer Review File · Nature Communications]

Reviewers' comments:

Reviewer #1 (Remarks to the Author):

This paper reports a study of $\text{CaBe}_2\text{P}_2\text{O}_8$ under high pressure by means of single crystal XRD. The authors determined the axial compressibility and equation of state for this compound. The authors reported phase transitions and a pressure driven coordination increase for Be and P. However, the increase of the coordination number in P from 4 to 6 has been already reported in *Nature Materials* (Ref. 18 of the manuscript). On the other hand, the authors claimed that six-fold coordination for an element of the second periodic is exotic, which indeed is not. For instance, in MgWO_4 ; Mg is six coordinated [*JOURNAL OF APPLIED PHYSICS* 107, 083506 (2010)] and it has been known for ninety years that Be is six coordinated in Be_2SiO_4 [*Proceedings of the Royal Society of London, Series A: Mathematical and Physical Sciences* 1927, 113, 642-657]. Therefore, I do not see much novelty in this work, in spite of the claims of the authors. The article could be a good work on HP crystallography, but this is a very well established technique, so there is no novelty on it. The only new result is the identification of the crystal structure of the HP phase. Basically, this is technical paper, with no new physics or chemistry on it. The reported data are useful for researchers in the area of minerals, but not important enough to warrant rapid publication in *Nature Communications*. My recommendation is to rewrite the article as a regular article including technical details and part of the Sup. Info. into the main body of the article and to submit the article either *Acta Cryst.* or *American Mineralogist*.

Reviewer #2 (Remarks to the Author):

Comments on the paper "Penta- and hexa-coordinated beryllium and phosphorous in high-pressure modifications of $\text{CaBe}_2\text{P}_2\text{O}_8$ " by Pakhomova et al.

The authors report about an experimental study where the first octal row element beryllium is shown to coordinate in an octahedral fashion. This is achieved by putting high pressure to the system. I think that the results are worth to be published in a prominent journal such as *Nature Communications*. However, the discussion and presentation of the work is done with outdated arguments and needs to be reconsidered. A revised version may become acceptable after the following points have been considered.

1. The authors repeatedly state that the higher coordination of atoms of second and higher octal row atoms is due to the availability of d-orbitals, which are not accessible by first octal row atoms. It has been shown already in the 1980s that the higher coordination of the heavier main group atoms is not due to the d-orbitals but due to the size of the atoms. I strongly recommend the authors to read the review article by Kutzelnigg (*Angew. Chem., Int. Ed. Engl.* 1984, 23, 272) and the theoretical work about the bonding analysis in heavier main group atoms such as *Aust. J. Chem.* 2004, 57, 1191. It has already been shown by Raoul Hoffmann in the 1970s that the higher coordination of heavier main-group atoms does not need d-orbitals at all. There are orbitals whose symmetry indicates that the ligand electrons do not engage in bonding with the metal atoms. Stabilization of these orbitals is due to the field effect of the metal atoms. A pertinent paper to this work is published in *J. Am. Chem. Soc.* 1972, 94, 3047. In fact, this has already been shown by Pimentel much earlier and was introduced in the literature as so-called three center two electron model (*J. Chem. Phys.* 1951, 19, 446).

2. Particular bonding properties and a change of the coordination of atoms in molecules under high pressures have been discussed in several works before. This should be acknowledged by citing proper references particular to the work by Roald Hoffmann, which can be found on his homepage. The work can be published after the above comments have been considered.

Reviewer #3 (Remarks to the Author):

The authors have carried out an interesting crystallographic study of high pressure polymorphs of $\text{CaBe}_2\text{P}_2\text{O}_8$, inspired by recent results for the isoelectronic material $\text{CaB}_2\text{Si}_2\text{O}_8$. They have demonstrated the occurrence of Be and P atoms in unusual 5- and 6-coordinated environments to oxygen. That result is of interest, since Be is typically usually only found in 4-coordinated environments in oxyanions. Overall, the report could be accepted for publication in *Nature Commun.* However, I believe that the discussion of the results in terms of "bonding" considerations is not at all sufficient, and it is presented in terms of rather vague and quite mundane statements about "covalent" vs "ionic" bonding. It could be made much more interesting and useful to a wider readership if it were developed more systematically and completely. In particular, I suggest that the authors could go much further in their discussion of what determines the packing arrangements around different atoms in crystals, and make a real advance in the field, if they consider the following.

First, the authors should think about what is happening as the crystals are compressed: the atoms will (a) attempt to maintain crystallographic translational symmetry, while (b) adjusting the overall packing scheme and hence individual coordination environments to achieve a more efficient use of space. Those packings are constrained by two factors: (i) local bonding energetics, balanced against (ii) non-bonded repulsive interactions. The present authors have focused only on the first of these: they need to consider the problem more holistically.

In the late 1970's/early 1980's, Mike O'Keeffe and Bruce Hyde developed the concepts of "eutaxy" and highlighted the relative roles of "bonding" between nearest neighbour atoms, balanced against repulsive "non-bonded" interactions among second-nearest neighbours, in determining the packing around all atomic species in a crystal. They also discussed in detail their reservations about the usual "ionic" bonding model for crystals, and showed that many crystal structure arrangements could be much better analysed by "eutaxy", considering the balance achieved between bonded and non-bonded interactions. I strongly suggest that the authors look at the two chapters written by Mike and Bruce in "Structure and Bonding In Crystals", edited by O'Keeffe and Navrotsky (Acad Press, 1981). They should also look at the arguments developed by these authors in *J Solid State Chem* 44 (1982) 24; *Nature* 309 (1984) 411. I believe that if they look at this work in detail, and consider the cooperative changes in ALL the atomic coordinations (i.e., Ca, Be, P and O; as well as for the corresponding compound containing B and Si), they will be able to present a much deeper understanding of what is actually occurring during densification of their crystals. This then goes towards a much better understanding of the factors that control atomic packing arrangements in crystals, that was always a key goal of O'Keeffe and Hyde.

Regarding the statements made about 5-coordinated Si, the authors have neglected to mention the initial report by Stebbins and McMillan of this unusual silicate coordination, for a silicate glass quenched from high pressure conditions (Am Mineral 74 (1989) 965). That was followed up by a more complete study, showing the presence of both 5- and 6-coordinated species as a function of glass composition and P conditions (Am Mineral 76 (1991) 8; Xue et al Science 245 (1989) 962). Interestingly, the 5-coordinated species were detectable in silicate glasses quenched from melts at ambient P conditions. Those results were interpreted using the O'Keeffe/Hyde models for bonded vs non-bonded repulsions and a local embodiment of the "eutaxy" principles (J Non-Cryst Solids 160 (1993) 116).

Reviewer 1

This paper reports a study of $\text{CaBe}_2\text{P}_2\text{O}_8$ under high pressure by means of single crystal XRD. The authors determined the axial compressibility and equation of state of this compound. The authors reported phase transitions and a pressure driven coordination increase for Be and P. However, the increase of the coordination number in P from 4 to 6 has been already reported in Nature Materials (Ref. 18 of the manuscript).

>> We were aware about the previous discoveries of the pressure-induced increase of P coordination number and have cited the corresponding literature in the reference list (References 11 and 18). Phosphorus is indeed known to be five- (ref.11) and six-fold (ref.18) coordinated in some high-pressure phases. However, our study obviously brings novelty into P high-pressure chemistry as soon as it is a first observation of step-wise increase of P coordination from tetrahedral (IV) to octahedral (VI) through the trigonal-pyramidal (V) intermediate state. In any case, main focus in our work is on the discovery of five- and six-fold coordinated Be. <<

On the other hand, the authors claimed that six-fold coordination for an element of the second periodic is exotic, which indeed is not. For instance, in MgWO_4 ; Mg is six coordinated [JOURNAL OF APPLIED PHYSICS 107, 083506 (2010)]

>> It is in agreement with common chemical nomenclature to address horizontal rows of the periodic table as *periods* and vertical columns as *groups*. Therefore by “elements of the second period” we mean Li, Be, B, C, N, O, F and Ne.

and it has been known for ninety years that Be is six coordinated in Be_2SiO_4 [Proceedings of the Royal Society of London, Series A: Mathematical and Physical Sciences 1927, 113, 642-657].

>> This comment is wrong as soon as Be is tetrahedrally coordinated in the crystal structure of Be_2SiO_4 . Structure of phenakite, Be_2SiO_4 , with BeO_4 tetrahedra was indeed proposed for the first time by Bragg in 1927 and proved in succeeding studies by Zachariasen (Kristallografiya (1971) 16, p1161-p1166) and by Kogure & Takeuchi (Mineralogical Journal (1986) 13, (1) p22-p27).

Therefore, I do not see much novelty in this work, in spite of the claims of the authors. The article could be a good work on HP crystallography, but this is a very well established technique, so there is no novelty on it. The only new result is the identification of the crystal structure of the HP phase. Basically, this is technical paper, with no new physics or chemistry on it. The reported data are useful for researchers in the area of minerals, but not important enough to warrant rapid publication in Nature Communications. My recommendation is to rewrite the article as a regular article including technical details and part of the Sup. Info. into the main body of the article and to submit the article either Acta Cryst. or American Mineralogist.

>> As we demonstrate above, Referee #1 criticism is groundless (*i.e.* confusing *periods* of the periodic table with *groups* and misleading idea of Be being six-fold coordinated in the crystal structure of Be_2SiO_4). In fact we report first case of progressive increase of phosphorus coordination number from four to five and to six, and for the first time

documented six-coordinated beryllium. In revised version of manuscript we described also results of theoretical calculations which support/explain our observations. Therefore our results are of general importance and novelty for inorganic chemistry, solid state physics, and material sciences.

Reviewer 2

Comments on the paper “Penta- and hexa-coordinated beryllium and phosphorous in high-pressure modifications of $\text{CaBe}_2\text{P}_2\text{O}_8$ ” by Pakhomova et al.

The authors report about an experimental study where the first octal row element beryllium is shown to coordinate in an octahedral fashion. This is achieved by putting high pressure to the system. I think that the results are worth to be published in a prominent journal such as *Nature Communications*. However, the discussion and presentation of the work is done with outdated arguments and needs to be reconsidered. A revised version may become acceptable after the following points have been considered.

1. The authors repeatedly state that the higher coordination of atoms of second and higher octal row atoms is due to the availability of *d*-orbitals, which are not accessible by first octal row atoms. It has been shown already in the 1980s that the higher coordination of the heavier main group atoms is not due to the *d*-orbitals but due to the size of the atoms. I strongly recommend the authors to read the review article by Kutzelnigg (*Angew. Chem., Int. Ed. Engl.* **1984**, 23, 272) and the theoretical work about the bonding analysis in heavier main group atoms such as *Aust. J. Chem.* **2004**, 57, 1191. It has already shown by Raoul Hoffmann in the 1970s that the higher coordination of heavier main-group atoms does not need *d*-orbitals at all. There are orbitals whose symmetry indicates that the ligand electrons do not engage in bonding with the metal atoms. Stabilization of these orbitals is due to the field effect of the metal atoms. A pertinent paper to this work is published in *J. Am. Chem. Soc.* **1972**, 94, 3047. In fact, this has already been shown by Pimentel much earlier and was introduced in the literature as so-called three center two electron model (*J. Chem. Phys.* **1951**, 19, 446).

>> We are very thankful to the reviewer for such a valuable comment that has motivated us to investigate the bonding question deeper and to perform DFT calculations. Study of the recommended literature taken together with the results of the *ab initio* calculations helped us to reconsider and to improve the discussion part. Our theoretical analysis suggests that the sequence of structural transitions of $\text{CaBe}_2\text{P}_2\text{O}_8$ is associated with the electronic transformations from predominantly molecular orbitals at low pressure to the state with overlapping electronic clouds of anions orbitals. We still have left little information on the Pauling's hybridization model: discussions on hybridisation model vs molecular orbital theory are still found in the literature and probably worthy to be addressed.

2. Particular bonding properties and a change of the coordination of atoms in molecules under high pressures have been discussed in several works before. This should be acknowledged by citing proper references particular to the work by Roald Hoffmann, which can be found on his homepage. The work can be published after the above comments have been considered.

>> We have taken into the account and cited several works by Roald Hoffmann that illuminate the question of the coordination number increase upon high-pressure

treatment. We are grateful for bringing these articles into our scope as they are valuable for the future work, especially the review article of Grocala et al (2007).

Reviewer 3

The authors have carried out an interesting crystallographic study of high pressure polymorphs of $\text{CaBe}_2\text{P}_2\text{O}_8$, inspired by recent results for the isoelectronic material $\text{CaB}_2\text{Si}_2\text{O}_8$. They have demonstrated the occurrence of Be and P atoms in unusual 5- and 6-coordinated environments to oxygen. That result is of interest, since Be is typically usually only found in 4-coordinated environments in oxyanions. Overall, the report could be accepted for publication in Nature Commun. However, I believe that the discussion of the results in terms of "bonding" considerations is not at all sufficient, and it is presented in terms of rather vague and quite mundane statements about "covalent" vs "ionic" bonding. It could be made much more interesting and useful to a wider readership if it were developed more systematically and completely.

>> In order to elaborate on the question of nature of pressure-induced changes we contacted group of Prof. I. Abrikosov and performed *ab initio* DFT calculations. The results of the calculations are presented in the revised manuscript and discussed. In the new version of the manuscript we avoid using "ionic" and "covalent" terms.

In particular, I suggest that the authors could go much further in their discussion of what determines the packing arrangements around different atoms in crystals, and make a real advance in the field, if they consider the following.

First, the authors should think about what is happening as the crystals are compressed: the atoms will (a) attempt to maintain crystallographic translational symmetry, while (b) adjusting the overall packing scheme and hence individual coordination environments to achieve a more efficient use of space. Those packings are constrained by two factors: (i) local bonding energetics, balanced against (ii) non-bonded repulsive interactions. The present authors have focused only on the first of these: they need to consider the problem more holistically.

In the late 1970's/early 1980's, Mike O'Keeffe and Bruce Hyde developed the concepts of "eutaxy" and highlighted the relative roles of "bonding" between nearest neighbour atoms, balanced against repulsive "non-bonded" interactions among second-nearest neighbours, in determining the packing around all atomic species in a crystal. They also discussed in detail their reservations about the usual "ionic" bonding model for crystals, and showed that many crystal structure arrangements could be much better analysed by "eutaxy", considering the balance achieved between bonded and non-bonded interactions. I strongly suggest that the authors look at the two chapters written by Mike and Bruce in "Structure and Bonding In Crystals", edited by O'Keeffe and Navrotsky (Acad Press, 1981). They should also look at the arguments developed by these authors in J Solid State Chem 44 (1982) 24; Nature 309 (1984) 411. I believe that if they look at this work in detail, and consider the cooperative changes in ALL the atomic coordinations (i.e., Ca, Be, P and O; as well as for the corresponding compound containing B and Si), they will be able to present a much deeper understanding of what is actually occurring during densification of their crystals. This then goes towards a much better understanding of the factors that control atomic packing arrangements in crystals, that was always a key goal of O'Keeffe and Hyde.

>> We are very thankful to the reviewer for bringing the "eutaxy" approach into our scope. However, after studying the recommended literature and trying to apply the

"eutaxy" approach to our case, we make a decision to describe overall crystal chemistry of hurlbutite (and other isotopological compounds of the $MT_1T_2O_8$ group) in traditional terms of "cationic coordination" and "anionic close packing". We did not observe a tendency for T cations to form a close packing while the high-pressure evolution of O and M atoms (M= Ca, Sr, Ba) toward the close packing is evident and describes smoothly the increasing coordination of T cations. We have added a brief crystal chemical discussion on HP behavior of $MT_1T_2O_8$ group to the manuscript. However, detailed and extended discussions of regularities in HP crystal chemistry of $MT_1T_2O_8$ compounds is out of scope of this work and deserve special paper in the crystallographic journal.

Regarding the statements made about 5-coordinated Si, the authors have neglected to mention the initial report by Stebbins and McMillan of this unusual silicate coordination, for a silicate glass quenched from high pressure conditions (Am Mineral 74 (1989) 965). That was followed up by a more complete study, showing the presence of both 5- and 6-coordinated species as a function of glass composition and P conditions (Am Mineral 76 (1991) 8; Xue et al Science 245 (1989) 962). Interestingly, the 5-coordinated species were detectable in silicate glasses quenched from melts at ambient P conditions. Those results were interpreted using the O'Keeffe/Hyde models for bonded vs non-bonded repulsions and a local embodiment of the "eutaxy" principles (J Non-Cryst Solids 160 (1993) 116).

>> We have added the missing references on 5- and 6-coordinated silicon in glasses and melts.

REVIEWERS' COMMENTS:

Reviewer #2 (Remarks to the Author):

The authors have answered my comments satisfactorily. I recommend acceptance of the revised manuscript.

Reviewer #3 (Remarks to the Author):

I think that the authors have adequately answered or addressed the questions raised by all reviewers. I am happy to recommend acceptance, if they correct the spelling of the name of Professor Mike O'Keeffe in the text.

REVIEWERS' COMMENTS:

Reviewer #2 (Remarks to the Author):

The authors have answered my comments satisfactorily. I recommend acceptance of the revised manuscript.

Reviewer #3 (Remarks to the Author):

I think that the authors have adequately answered or addressed the questions raised by all reviewers. I am happy to recommend acceptance, if they correct the spelling of the name of Professor Mike O'Keeffe in the text.

The spelling of Professor Mike O'Keeffe is changed.